# CDG-MAE: Learning Correspondences from Diffusion Generated Views

## Abstract

Dense correspondences are critical for applications such as video label propagation, but learning them is hard because of tedious and unscalable manual annotation needs. Self-supervised methods address this by using a cross-view pretext task, often modeled with a masked autoencoder, where a masked target view is reconstructed from an anchor view. However, acquiring effective training data remains a challenge - collecting diverse video datasets is costly, while simple image crops lack the necessary pose variations, underperforming video-based methods. This paper introduces CDG-MAE, a novel MAE-based self-supervised method that uses diverse synthetic views generated from static images via an image-conditioned diffusion model. We present a quantitative method to evaluate the local and global consistency of the generated views to choose the right diffusion model for cross-view self-supervised pretraining. These generated views exhibit substantial changes in pose and perspective, providing a rich training signal that overcomes the limitations of video and crop-based anchors. Furthermore, we enhance the standard single-anchor MAE setting to a multi-anchor masking strategy to increase the difficulty of the pretext task. CDG-MAE substantially narrows the gap to video-based MAE methods, while maintaining the data advantages of image-only MAEs.

## 1 Introduction

Masked Autoencoders (MAEs) learn rich visual representations by reconstructing randomly masked parts of an image from the remaining visible context (He et al., 2022). The paradigm of learning by reconstruction naturally extends to multi-view scenarios through cross-view correspondence learning (Weinzaepfel et al., 2022; 2023; Gupta et al., 2023; Eymaël et al., 2024). These methods exploit the redundancy in captured information and the inherent 3D consistency across viewpoints as strong cues for learning to model dynamics, physics and semantics. A specific adaptation, cross-view masked auto-encoding, tasks a model to reconstruct a masked view of a scene from another anchor view. By learning to complete missing parts of scene representations, cross-view masked auto-encoding leads to strong vision models capable of understanding underlying scene semantics.

Training vision models to learn correspondences requires capturing multiple images of the scene, which in the real world can be costly. A common shortcut for *static environments* uses simulators to render diverse views of a scene (Weinzaepfel et al., 2022; 2023). To model motion and perspective changes, the data itself must exhibit dynamic changes. Collecting videos is a good alternative (Gupta et al., 2023), but this comes with an acquisition cost as well as the more limited diversity of the scenes one can capture. For example, a video captures only a single motion scenario in a scene.

Given the large availability of 2D images, can we generate dynamic variations from images equivalent to those found in videos for correspondence learning? A simple approach is to emulate changes with augmentations such as image crops (Eymaël et al., 2024), but the cropped view diversity is limited. As acknowledged by Eymaël et al. (2024), crops cannot introduce variations in pose, limiting the richness of learned inductive biases. Consequently, there is a need to develop methods that can use static images to derive richer, pose-variant transformations found in real-world dynamic scenes.

Diffusion models perform well in image generation (Dhariwal & Nichol, 2021; Labs, 2024), employing various conditioning mechanisms to guide the generation process (Rombach et al., 2022; Zhang et al., 2023). Conditioning diffusion models with image embeddings enables the generation of

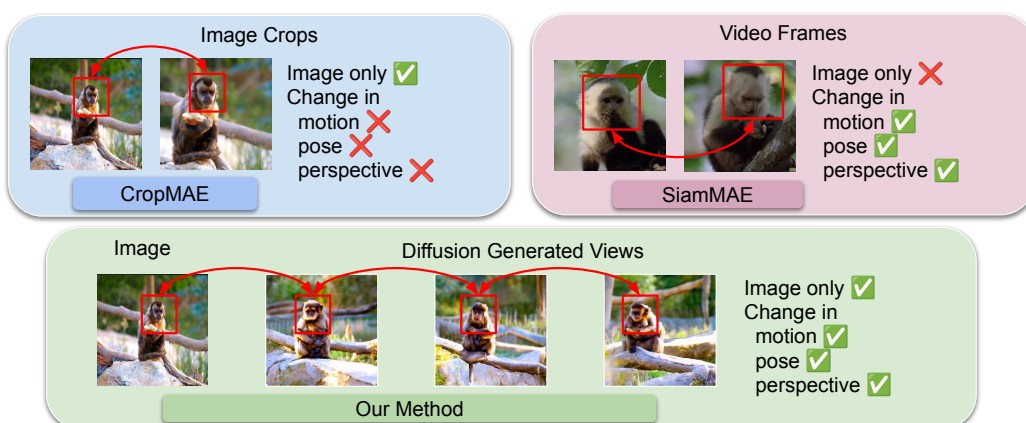

Figure 1: **CDG-MAE**: We train a vision encoder in a self-supervised manner by finding correspondences between real and synthetic views generated by a diffusion model ( Belagali et al. (2024)). These synthetic views preserve important scene information while introducing diverse dynamics.

diverse variations of an input image in a self-supervised way (S-LDMs Li et al. (2024); Belagali et al. (2024); Ma et al. (2025)). Crucially, we observe that these variations can introduce different perspectives or diverse motions, equivalent to individual video frames (see Figure 1). However, in order to learn correspondences, the generated views should introduce local changes while maintaining global consistency. Currently, there are no quantitative tools to evaluate such properties.

We introduce *CDG-MAE*, the first cross-view self-supervised learning method for correspondence learning, to train Masked Autoencoders using views generated from a diffusion model. However, the success of this strategy depends on the careful selection of the diffusion model as not all diffusion-generated views are useful for correspondence learning. In order for such views to provide a training signal analogous to video data, they must exhibit meaningful changes in pose and image location. To ensure this, we introduce quantitative consistency metrics to guide us in choosing the right diffusion model for generating these views. We observe a strong effect of these metrics on final performance. Finally, we extend the current single-anchor-view setting in cross-view self-supervised learning by using additional anchor views and applying an anchor-specific masking strategy to increase the difficulty of the pretext task. Our contributions are as follows:

**(i) Diffusion-based view generation for MAE training.** We are the first to explore training cross-view correspondence MAEs using diffusion-generated views to address the limitations of video and image-crop based cross-view MAE methods.

**(ii) A method to evaluate the utility of diffusion generated-views for correspondence learning.** We develop quantitative metrics to evaluate local and global consistency between views. We demonstrate their effectiveness in choosing the right diffusion model for cross-view self-supervised learning.

**(iii) Multi-anchor masking as a novel MAE training paradigm.** We extend the standard single-anchor MAE setting to a multi-anchor framework. Having multiple anchors allows for anchor masking, which creates a more challenging and effective pretext task.

We show that CDG-MAE, trained with diffusion-generated data and our multi-anchor setting, achieves substantial improvements over state-of-the-art MAE methods reliant on image crops and narrows the performance gap with video-based approaches.

## 2 RELATED WORK

**Self-supervised learning** — Masked Image Modeling (MIM) is a self-supervised learning paradigm that masks part of the input visual data and trains models to predict the masked parts using visible parts (He et al., 2022; Bao et al., 2021; Xie et al., 2022; Wei et al., 2022; Tong et al., 2022; Assran et al., 2023). Specifically, Masked Autoencoders (MAE He et al. (2022)) divide an image into

patches, and mask some of them. An encoder extracts features from visible patches only. The encoder features and appended mask tokens with positional encoding are used to decode the patch pixel values. With a sufficiently high masking ratio, the encoder learns robust visual features for downstream discriminative tasks (classification, segmentation, object detection). VideoMAE (Tong et al., 2022) pretrains on videos integrating multiple frames. Another class of SSL methods are view-invariant methods which use two augmentations of the same image, and train the model to match the global/local features between augmentations (Chen et al., 2020; Caron et al., 2021; Grill et al., 2020; He et al., 2020; Zhou et al., 2021; Oquab et al., 2023; Bardes et al., 2022). In this work, we specifically focus on MAE as a SSL framework due to its efficiency and modularity.

**Cross-view self-supervised learning** — learns visual features that match cross-views either for video (Gupta et al., 2023; Eymaël et al., 2024) or 3D (Weinzaepfel et al., 2022; 2023) downstream tasks. These works use Siamese Masked Autoencoders to learn cross-view correspondences. Pretraining employs two views: anchor and target. The masked target image passes through the encoder, then the decoder reconstructs the masked patches. The anchor view passes through the encoder independently, without masking. To facilitate cross-view learning, the decoder reconstructs the target view by cross-attending to the encoder features of the anchor view. A high masking ratio forces the encoder to learn features that match patches of the anchor view to the target view. Siamese Masked Autoencoders (SiamMAE Gupta et al. (2023)) extract target and anchor as two different video frames. Given object motion, view point change, and pose change in video, SiamMAE visual features are suitable for label propagation downstream tasks: video object propagation (Pont-Tuset et al., 2017a), semantic part propagation (Zhou et al., 2018a), and pose propagation (Jhuang et al., 2013a). Cropped Siamese Masked Autoencoder (CropMAE Eymaël et al. (2024)) extends SiamMAE, extracting the two views from two crops of the same image, obviating the need of video pretraining. CropMAE performs worse on tasks like pose propagation, as the anchor and target views have limited pose changes.

**Diffusion models** — generate realistic images due to breakthroughs in conditioning (Ho & Salimans, 2022; Zhang et al., 2023), architecture (Peebles & Xie, 2023; Esser et al., 2024) and sampling (Song et al., 2020; Lu et al., 2023). Latent Diffusion Models (LDMs Rombach et al. (2022)) efficiently train a diffusion model in a compact VAE latent space instead of pixel space. While diffusion models are commonly conditioned on explicit signals such as class labels and text captions, recent works (Belagali et al., 2024; Li et al., 2024; Ma et al., 2025; Graikos et al., 2024) train conditional diffusion models in a self-supervised way (S-LDMs). These approaches first train an image encoder using view-invariant self-supervised learning methods ( Caron et al. (2021); Chen et al. (2021)) to learn image embeddings. The diffusion model is then conditioned on the output of the frozen encoder.

Diffusion models are increasingly used for data augmentation, including self-supervised settings. Tian et al. (2024a;b) use Stable Diffusion to generate augmentations, whereas, approaches like Gen-SIS (Belagali et al., 2024) train the diffusion model on the same dataset used for the main SSL task. We experiment with three different self-supervised image-conditioned diffusion models: Gen-SIS (Belagali et al., 2024), RCG (Li et al., 2024), and Lumos (Ma et al., 2025). Gen-SIS and RCG are pretrained on ImageNet-1K, while Lumos is pretrained on 190M open-source images.

## 3 METHOD

Our approach, CDG-MAE, consists of three stages: 1) Bag of views generation. 2) Quantitative view evaluation and 3) Cross-view MAE training. Figure 2 describes the overall pipeline of CDG-MAE.

### 3.1 BAG OF VIEWS

Given an image-only dataset (e.g., ImageNet), we generate $M$ alternative views of the scene depicted by each real image via an off-the-shelf image-conditioned diffusion model. We mainly use a Self-supervised Latent Diffusion Model (S-LDM), pretrained with ImageNet self-supervision (Belagali et al., 2024). S-LDM follows the Latent Diffusion Model (Rombach et al., 2022) architecture and is conditioned with an image encoder pretrained with view invariant SSL (Caron et al., 2021) and frozen during LDM training. We observe that S-LDM generates diverse views of an input image with motion, pose, and perspective variations, mimicking changes between video frames (Figure 1). These offline generated views, along with the original image, are the *bag of views* for that input, avoiding online sampling during MAE training (with a one-time cost of 135 ms per real image).

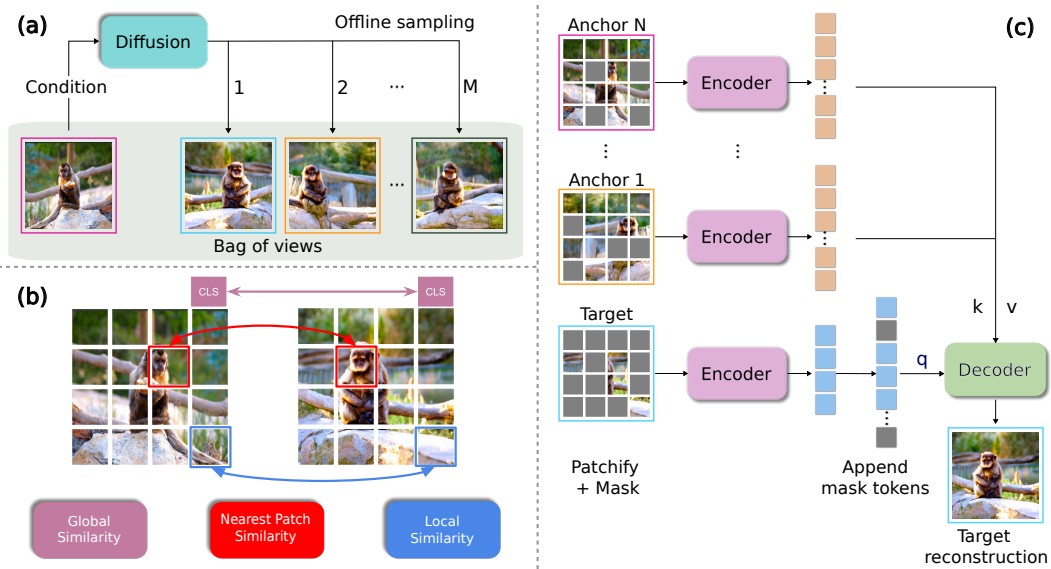

Figure 2: **Overview of CDG-MAE**: **(a)** For every real image, we generate $M$ views using an off-the-shelf S-LDM (Belagali et al., 2024). **(b)** We develop quantitative metrics to evaluate local and global consistencies between view pairs. **(c)** We develop a multi-anchor framework to train cross-view MAE. Having multiple anchors allows for anchor masking, which creates a more challenging pretext task.

## 3.2 EVALUATING CONSISTENCY BETWEEN VIEW PAIRS

Intuitively, an ideal pair of views $(V_1, V_2)$ for correspondence learning should feature the same set of objects undergoing transformations in motion, pose, and perspective. Such views must therefore exhibit local variations that reflect these transformations while maintaining global consistency. To measure these properties, we developed consistency metrics. Figure 2 (b) provides a visual illustration.

Let $f_*(\cdot)$ and $f_i(\cdot)$ denote functions that extract a transformer-based global and local embedding at a spatial location $i$, respectively for a single input $V_1$. In our case, $f_*(V_1)$ is the $[CLS]$ token and $f_i(V_1)$ is the $i^{th}$ patch token extracted from a ViT encoder. We use the pretrained ViT-B/16 MAE (He et al., 2022) to extract both patch tokens and CLS tokens. Let us assume there are $L$ distinct spatial locations, indexed from 1 to $L$. Using this information, we calculate the:

**Global Similarity (GS)** — Measures the overall semantic coherence between views $V_1$ and $V_2$. It is defined as the cosine similarity ($sim(\mathbf{u}, \mathbf{v})$) between their respective global embeddings, $f_*(V_1)$ and $f_*(V_2)$. High GS is desirable, indicating that global semantic content is preserved.

**Local Similarity (LS)** — It is the average cosine similarity between local embeddings $f_i(V_1)$ and $f_i(V_2)$ from $V_1$ and $V_2$ respectively, at each identical spatial location $i$. A low LS indicates change in motion, pose, and perspective between views.

**Nearest Patch Similarity (NPS)** — Provides a measure of global consistency, especially in the presence of transformations. For each local embedding $f_i(V_1)$ in $V_1$, we identify its nearest neighbor (most similar) among all local embeddings $\{f_j(V_2)\}_{j=1}^{L}$ from $V_2$. NPS is then calculated as the average of these maximum similarity scores across all $L$ locations in $V_1$. Even with significant changes in motion, pose, or perspective, a high NPS is expected if the views remain globally coherent.

$$GS(V_1, V_2) = sim(f_*(V_1), f_*(V_2)) \tag{1}$$

$$LS(V_1, V_2) = \frac{1}{L} \sum_{i=1}^{L} sim(f_i(V_1), f_i(V_2)) \tag{2}$$

$$NPS(V_1, V_2) = \frac{1}{L} \sum_{i=1}^{L} \left( \max_{j \in \{1, \dots, L\}} sim(f_i(V_1), f_j(V_2)) \right) \tag{3}$$

Table 1 presents the evaluation of quantitative metrics on several types of view pairings: (i) video frames, (ii) synthetic views generated by the S-LDM, (iii) k-nearest neighbor (k-nn) image pairs from the training data (derived from cosine similarity in S-LDM conditioning encoder space), and (iv) random image pairs. The result demonstrates that S-LDM generated views are much closer to video frames than k-nn images or random pairs. This indicates that S-LDM, conditioned on static real images, can mimic characteristics of video data, making it ideal for correspondence learning. In Section 5.2, we demonstrate the effectiveness of the above metrics in choosing the right diffusion model for view generation.

Table 1: Quantitative evaluation of diffusion generated views when compared to video frames, k-nn images, and random pair of images. We sample random 5000 images from ImageNet (Deng et al., 2009) and 5000 pair of video frames from Kinetics-400 (Kay et al., 2017) for calculation.

| Views | Global Sim. ($\uparrow$) | Local Sim. ($\downarrow$) | Nearest Patch Sim.($\uparrow$) |
|---|---|---|---|
| Video frames | 0.992 | 0.389 | 0.884 |
| Diffusion (S-LDM) | 0.992 | 0.377 | 0.795 |
| K-nn images | 0.951 | 0.301 | 0.719 |
| Random images | 0.892 | 0.175 | 0.600 |

## 3.3 CDG-MAE OVERALL DESIGN AND TRAINING STRATEGY

In this section, we explain the overall design and training methodology for CDG-MAE: cross-view masked autoencoders using diffusion generated views. Consistent with existing works on cross-view MAE (Gupta et al., 2023; Eymaël et al., 2024), the pretext task is the reconstruction of randomly masked patches in a target, using visible target patches and anchor views. The architecture is an encoder-decoder Vision Transformer (ViT), where the target and each anchor are independently processed by a weight-shared ViT encoder. Subsequently, the decoder appends mask tokens to the visible target tokens, and reconstructs the content for these masked patches. This reconstruction is conditioned on visible target tokens through self-attention and on anchor tokens via cross-attention.

In the remainder of this section, we denote the encoder and decoder as $e_\theta$ and $d_\psi$ respectively. We define image patchification operator as $\rho(\cdot)$, masking operator as $m(\cdot, \text{ratio})$, and concatenation as $[\cdot; \cdot]$. The masking ratios for the target and anchor views are denoted by $r_t$ and $r_a$, respectively.

**Encoding Target** — The target image $T \in \mathbb{R}^{H \times W \times 3}$ is first patchified into a sequence of $N_t = (H/P) \times (W/P)$ non-overlapping patches, each of size $P \times P \times 3$. We then flatten these patches into a 1D sequence, and apply random masking using a high target masking ratio ($r_t$). We discard the masked patches $\tilde{T}_v$, and process the visible patches $T_v$ through the encoder $e_\theta$ to obtain encoder target representations $T'_v$.

$$T_p = \rho(T), \quad T_v, \tilde{T}_v = m(T_p, r_t) \tag{4}$$
$$T'_v = e_\theta(m(T_p, r_t); \theta) \tag{5}$$

**Multi-anchor and anchor-masking** — Traditional cross-view MAEs such as SiamMAE (Gupta et al., 2023) and CropMAE (Eymaël et al., 2024) typically encode a single unmasked anchor view. In CDG-MAE, we propose leveraging multiple anchor views, $\{A^k\}_{k=1}^N$, sampled from the "bag of views" (where $N$ is the number of anchors). Furthermore, we introduce *anchor masking*: each anchor view $A^k$ is independently masked with a specific anchor masking ratio ($r_a$), allowing for fine-grained control over the difficulty of the pretext task. Higher $N$ can provide the decoder with richer contextual information, simplifying target reconstruction. Conversely, applying anchor masking makes the task more challenging by reducing the visible information from each anchor. We demonstrate that an optimal balance between $N$ and $r_a$ enhances representation learning.

Similar to target encoding, each anchor $A^k$ is patchified and masked. We discard masked anchor patches, and pass the visible patches through the weight-shared encoder $e_\theta$ (Siamese-style encoding) to obtain anchor tokens $A'^k_v$. We then concatenate the output tokens from all $N$ anchors to form a aggregated anchor representation $A'_v$

$$A_v'^k = e_\theta(m(\rho(A^k), r_a); \theta), \qquad\qquad \forall k \in \{1, \dots, N\} \qquad (6)$$

$$A_v' = [A_v'^1; A_v'^2; ..; A_v'^N] \qquad (7)$$

**Target reconstruction** — The input to the decoder $d_\psi$ is the sequence $T_a$, which is a concatenation of encoder target representations $T_v'$ and mask tokens $M_{\tilde{T}_v}$. The decoder self-attends to all tokens within $T_a$, and cross-attends to the aggregated anchor representation $A_v'$, allowing it to leverage information across anchor views to predict the masked target patches.

Our multi-anchor setting encourages the encoder $e_\theta$ to learn features that are robust for matching across a diverse set of views - beyond just two views as in prior work (Gupta et al., 2023; Eymaël et al., 2024). We apply a reconstruction loss (MSE) between masked target patches $\tilde{T}_v$ and decoder predictions $T_r$, following prior work.

$$T_a = [T_v', M_{\tilde{T}_v}] \qquad (8)$$

$$T_r = d_\psi(T_a, A_v'; \psi) \qquad (9)$$

$$\mathcal{L}(T_r, T_p) = \frac{1}{|\tilde{T}_v|} \left\| T_r - \tilde{T}_v \right\|_2^2 \qquad (10)$$

## 4 EXPERIMENTAL SETTING

**Bag of Views creation** — For each real image in the ImageNet-1K (Deng et al., 2009) training dataset, we generate $M = 4$ random synthetic views using the pretrained checkpoint of S-LDM (Belagali et al., 2024). The real image along with generated views are treated as the *bag of views*. The generation is done in offline mode and stored on the disk before training CDG-MAE. Following (Belagali et al., 2024), we use a classifier-free guidance weight (Ho & Salimans, 2022) of 6 and 50 DDIM (Song et al., 2020) steps for sampling.

**Training** — We utilize the official codebase of CropMAE (Eymaël et al., 2024) and closely follow their setting. By default, we use a ViT-S/16 encoder and a four-layer decoder. Each decoder block has an embedding dimension of 256, and contains cross-attention, feedfoward and self-attention modules. We train for 100 epochs on ImageNet-1K with a base learning rate of $1.5 \times 10^{-4}$ and batch size of 2048.

From the *bag of views* (containing $M$ views), one image is randomly chosen as the target and $N$ additional images as anchors ($N < M$). We use a target masking ratio $r_a = 90\%$. In the multi-anchor setting, we apply uniform anchor masking ratio across all anchors, with each anchor masked independently and randomly. We also investigate the impact of training with a reduced patch size by training both CropMAE and CDG-MAE with a ViT-S/8 backbone for 100 epochs. More training details are provided in Appendix A.1.1.

**Downstream evaluation** — Following existing works (Gupta et al., 2023; Eymaël et al., 2024) we evaluate correspondence learning using three label propagation tasks in videos - 1) DAVIS-2017 video object segmentation (Pont-Tuset et al., 2017b), 2) VIP semantic part propagation (Zhou et al., 2018b), and 3) JHMDB human pose propagation (Jhuang et al., 2013b). In label propagation, we are provided with the annotation of the first frame and the task is to propagate the label to all frames by computing the similarity (correspondence) between patches of frames. The evaluation is done in a training-free manner using the pretrained encoder following the setting of (Gupta et al., 2023; Eymaël et al., 2024). More details are provided in the Appendix A.1.2.

## 5 RESULTS

We first discuss the design choices of CDG-MAE in Sec 5.1,5.2, 5.3 and then compare with MAE-based methods in Sec 5.4. We present results on training with a smaller patch size in Sec 5.5.

Table 2: It is optimal to choose real or generated image as the target view.

| Target View | DAVIS $\mathcal{J}\&\mathcal{F}_m$ | VIP mIoU | JHMDB PCK0.1 |
|---|---|---|---|
| Always Real | 61.3 | 37.1 | 46.8 |
| Always Generated | 60.0 | 37.1 | 46.5 |
| Real or Generated | 61.2 | 37.6 | 46.5 |
| k-nn image | 60.5 | 36.0 | 46.5 |

Table 3: A balanced target masking ratio ($r_t = 90\%$) yields best performance.

| Masking Ratio (%) | DAVIS $\mathcal{J}\&\mathcal{F}_m$ | VIP mIoU | JHMDB PCK0.1 |
|---|---|---|---|
| 75 | 60.0 | 34.9 | 46.2 |
| 90 | 61.2 | 37.6 | 46.5 |
| 98.5 | 60.7 | 35.5 | 44.4 |

## 5.1 TARGET SELECTION AND MASKING

To investigate the influence of target selection, we employ a simple single-anchor configuration where the anchor view is unmasked ($r_a = 0$) and with a high target masking ratio ($r_t = 90\%$). We evaluate three strategies for selecting the target view: 1) always using the real image, 2) always using a diffusion-generated view or 3) randomly choosing between real and generated. We observe in Table 2 that always using the real image as target and the random choice strategy yield comparable, strong performance. Hence, we adopt the random choice selection as the default strategy for CDG-MAE.

In Table 2, we also tested using k-nearest neighbor ($k = 5$) image pairs for anchor and target, and observe that it underperforms our default strategy. While k-nn images might share global feature similarity, Table 1 indicates they lack the high Nearest Patch Similarity found in video frames.

Next, in Table 3, we investigate the effect of target masking ratio ($r_t$). Unlike vanilla MAE (which uses $r_t = 75\%$), cross-view MAEs such as (Eymaël et al., 2024; Gupta et al., 2023) typically employ higher ratios ($\geq 90\%$). Lower ratios (e.g., 75%) can encourage the target to reconstruct itself, thereby hindering correspondence learning from anchor views. CropMAE uses a very high ratio ($r_t = 98.5\%$) to keep the task challenging under high information redundancy between anchor and target cropped from the same image. In CDG-MAE, such high ratio performs poorly. This can be attributed to the greater visual variations between diffusion generated anchor views and the target image compared to simple image crops. In our case, target reconstruction requires more context. We observe that a balanced masking ratio of $r_t = 90\%$ yields optimal performance.

## 5.2 DIFFUSION MODEL SELECTION

In this section, we experiment training CDG-MAE (single-anchor setting) using views generated from three diffusion models pretrained in a self-supervised manner (S-LDM): Gen-SIS (Belagali et al., 2024), RCG (Li et al., 2024), and Lumos (Ma et al., 2025). Gen-SIS and RCG are trained on ImageNet-1K, while Lumos is trained on a large collection of 190M images from open source datasets. Table 4 shows the performance of CDG-MAE when trained with views from different diffusion models, along with our proposed consistency metrics (GS, LS, NPS). We also report the metrics on Kinetics video frames for reference. The table shows that views generated from Gen-SIS and Lumos are considerably closer to video frames than RCG-generated views in terms of GS, LS, and NPS. Similarly, we observe that training with views from either Gen-SIS or Lumos offers higher performance than training with RCG. This finding demonstrates the importance of our proposed metrics in identifying the right diffusion model for learning correspondences. Based on these metrics, we chose Gen-SIS over RCG as our default diffusion model. Since Lumos is trained on a larger scale dataset than ImageNet-1K, we do not use it in our experiments to avoid potential data leakage effects. We did not include image-to-video diffusion models in our study because it would require immense computational resources (tens of thousands of A100 hours) to generate videos at ImageNet scale (see Appendix A.4.4).

## 5.3 MULTI-ANCHOR AND ANCHOR-MASKING

In this section, we explore the extension from a single-anchor to a multi-anchor framework. The latter enables us to introduce anchor masking, a technique not explored in prior methods such as SiamMAE (Gupta et al., 2023) and CropMAE (Eymaël et al., 2024).

Table 4: CDG-MAE performance with different diffusion models (S-LDMs) and corresponding consistency metrics (GS, LS, NPS). Video frame metrics provided as reference. Our proposed consistency metrics strongly effect the performance.

| Diffusion Model | DAVIS | VIP | JHMDB | Global Sim. ($\uparrow$) | Local Sim. ($\downarrow$) | Nearest Patch Sim.($\uparrow$) |
|---|---|---|---|---|---|---|
| Gen-SIS ( Belagali et al. (2024)) | 61.2 | 37.6 | 46.5 | 0.992 | 0.377 | 0.795 |
| RCG ( Li et al. (2024)) | 57.4 | 34.8 | 43.7 | 0.955 | 0.308 | 0.738 |
| Lumos ( Ma et al. (2025)) | 61.9 | 37.7 | 47.3 | 0.995 | 0.376 | 0.812 |
| Video frames | NA | NA | NA | 0.992 | 0.389 | 0.884 |

Table 5 demonstrates that transitioning from a single anchor to two anchors substantially improves performance. With two anchors, the decoder can learn more robust correspondences by matching target patches across both views. Within the two-anchor configuration, applying an anchor masking ratio of 25% or 50% further enhances the results. Ultimately, we achieve optimal performance using three anchors with a 25% anchor masking ratio, as presented in Table 5. This highlights the need for anchor masking when increasing the number of anchors to avoid providing too much information to the decoder, which would greatly reduce the task difficulty.

Table 5: Effect of multiple anchors and anchor masking ($r_a$). Multi-anchor training improves performance, and anchor masking offers control over pretext task difficulty. We report mean $\pm$ std across 3 pretraining runs (seeds).

| Num. of Anchors ($N$) | Anchor Masking ratio ($r_a$) | DAVIS $\mathcal{J}\&\mathcal{F}_m$ | VIP mIoU | JHMDB PCK0.1 |
|---|---|---|---|---|
| 1 | 0 | 61.2$\pm$0.0 | 37.6 $\pm$0.4 | 46.5 $\pm$0.3 |
| 2 | 0 | 62.0 $\pm$0.1 | 37.6 $\pm$0.1 | 47.1 $\pm$0.2 |
| 2 | 25% | 62.4 $\pm$0.2 | 38.0 $\pm$0.3 | 47.3 $\pm$0.1 |
| 2 | 50% | 62.1 $\pm$0.1 | 38.1 $\pm$0.2 | 47.8 $\pm$0.1 |
| 3 | 25% | 62.6 $\pm$0.1 | 38.1 $\pm$0.1 | 47.8 $\pm$0.2 |
| 3 | 50% | 62.0 $\pm$0.4 | 37.4 $\pm$0.3 | 47.5 $\pm$0.2 |

## 5.4 COMPARISON WITH OTHER MASKED AUTOENCODERS

We compare CDG-MAE with multiple MAE baselines. These include the vanilla MAE (He et al., 2022), video-based MAEs - Video-MAE (Tong et al., 2022) and MAE-ST (Feichtenhofer et al., 2022), and other cross-view MAEs such as CropMAE (Eymaël et al., 2024), SiamMAE (Gupta et al., 2023), and CroCo (Weinzaepfel et al., 2022; 2023). We evaluate two CDG-MAE variants: CDG-MAE-a1 (single unmasked anchor) and CDG-MAE-a3 (three anchors with 25% masking).

Results in Table 6 show that CDG-MAE substantially outperforms CropMAE across all downstream tasks, indicating the effectiveness of using diffusion generated views over crops of an image. Furthermore, CDG-MAE-a3 achieves better performance than SiamMAE across most metrics. This is noteworthy because CDG-MAE is pretrained on the ImageNet-1K dataset, whereas SiamMAE is trained with video frames from Kinetics-400 dataset, and downstream evaluation tasks are video-based. This finding demonstrates the efficacy of our multi-anchor and anchor masking strategy for learning correspondences from static images.

## 5.5 IMPACT OF FINER GRAINED REPRESENTATIONS

SiamMAE has demonstrated that reducing the encoder patch size from 16 to 8 can substantially enhance performance. At patch size 8, SiamMAE can learn more robust correspondences from fine-grained changes (variations of small objects) between two frames. We investigate the effect of smaller patch size for CDG-MAE, and compare with SiamMAE and CropMAE by training a ViT-S/8 encoder. Results are presented in Figure 3.

CropMAE shows no performance improvement with ViT-S/8 in object and part propagation. This is likely because cropped views lack sufficient fine-grained variations to benefit smaller patch training. In contrast, both SiamMAE and CDG-MAE exhibit notable performance gains with ViT-S/8 across all three tasks, compared to their ViT-S/16 variants. CDG-MAE substantially outperforms image-based

Table 6: Comparison of CDG-MAE with other MAE based methods. a1 and a3 refer to single anchor and three anchors with 25% anchor masking respectively. † refers to our reproduction on ImageNet-1K. The best and second-best results are highlighted in **Bold** and Underline.

| Method | Arch | Dataset | Epochs | DAVIS | | | VIP | JHMDB | |
| --- | --- | --- | --- | --- | --- | --- | --- | --- | --- |
| | | | | $\mathcal{J}\&\mathcal{F}_m$ | $\mathcal{J}_m$ | $\mathcal{F}_m$ | mIoU | PCK0.1 | PCK0.2 |
| MAE ( He et al. (2022)) | ViT-B/16 | ImagNet-1K | 1600 | 53.5 | 52.1 | 55.0 | 28.1 | 44.6 | 73.4 |
| Video-MAE ( Tong et al. (2022)) | ViT-S/16 | Kinetics-400 | 800 | 39.3 | 39.7 | 38.9 | 23.3 | 41.0 | 67.9 |
| MAE-ST ( Feichtenhofer et al. (2022)) | ViT-L/16 | Kinetics-400 | 800 | 54.6 | 55.5 | 53.6 | 33.2 | 44.4 | 72.5 |
| CroCov1 ( Weinzaepfel et al. (2022)) | ViT-B/16 | Habitat | 400 | 55.9 | 52.9 | 58.9 | 31.3 | 42.3 | 70.6 |
| CroCov2 ( Weinzaepfel et al. (2023)) | ViT-B/16 | Habitat + Real | 100 | 56.5 | 53.0 | 60.0 | 32.1 | 44.6 | 72.8 |
| CroCov2 ( Weinzaepfel et al. (2023)) | ViT-L/16 | Habitat + Real | 100 | 57.9 | 54.4 | 61.4 | 31.7 | 43.4 | 71.3 |
| SiamMAE ( Gupta et al. (2023)) | ViT-S/16 | Kinetics-400 | 2000 | 62.0 | **60.3** | 63.7 | 37.3 | 47.0 | 76.1 |
| CropMAE † ( Eymaël et al. (2024)) | ViT-S/16 | ImagNet-1K | 100 | 59.7 | 56.9 | 62.5 | 33.8 | 43.9 | 72.3 |
| CDG-MAE-a1 | ViT-S/16 | ImagNet-1K | 100 | 61.2 | 58.1 | 64.3 | 37.6 | 46.5 | 75.5 |
| CDG-MAE-a3 | ViT-S/16 | ImagNet-1K | 100 | **62.6** | 59.7 | **65.5** | **38.1** | **47.8** | **76.3** |

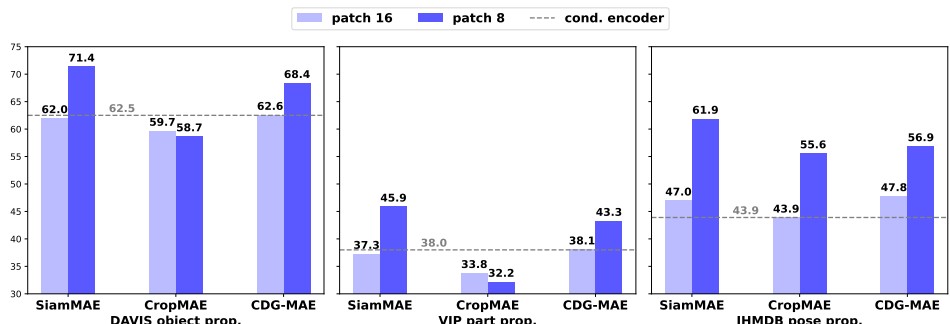

Figure 3: Performance of SiamMAE, CropMAE and CDG-MAE with ViT-S/16 and ViT-S/8 on DAVIS ($\mathcal{J}\&\mathcal{F}_m$), VIP (mIoU), and JHMDB (PCK0.1). We also present the performance of S-LDM's conditioning encoder with a dashed line.

CropMAE and closes the gap to the video-based SiamMAE. This highlights the ability of diffusion-generated views to provide diverse and rich variations beneficial for correspondence learning.

Furthermore, we evaluate the conditioning encoder of S-LDM (Belagali et al., 2024) on these downstream tasks. As indicated by the dashed line in Figure 3, while this encoder performs adequately on object and part propagation when compared to our ViT-S/16 encoder, it substantially underperforms in pose propagation. CDG-MAE ViT-S/8 outperforms the conditioning encoder in all three tasks. This shows that scaling tokens (patch size reduction) allows to better leverage diffusion-generated data, outperforming the original encoder used to condition the generation of such data.

## 6 CONCLUSION

We introduced CDG-MAE, a novel MAE framework for learning cross-view correspondence using diffusion-generated views. We developed new metrics to evaluate the local and global consistency of generated views. Such properties, inherent in video data, are important to learn correspondences. We demonstrate the effectiveness of the proposed consistency metrics in choosing the right diffusion model for view generation. CDG-MAE, trained with diffusion views derived from static images along with our proposed multi-anchor masking, substantially outperforms existing crop-based MAE methods and narrows the performance gap with video-based approaches. We hope our work inspires further exploration into synthetic data generation to leverage rich and diverse image datasets for cross-view representation learning.

**Reproducibility Statement** — We have provided implementation details in Sections 4 and A.1. The code will be released upon publication.

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

# A  APPENDIX

## A.1  IMPLEMENTATION DETAILS

### A.1.1  TRAINING

Our implementation is built using the CropMAE (Eymaël et al., 2024) codebase. We use ImageNet-1K as our pretraining dataset. The default training hyperparameters for our method (CDG-MAE) under the single anchor setting are presented in Table 7. We also provide the hyperparameters of CropMAE as a reference. CDG-MAE also uses cropping as an augmentation, following SiamMAE (Gupta et al., 2023).

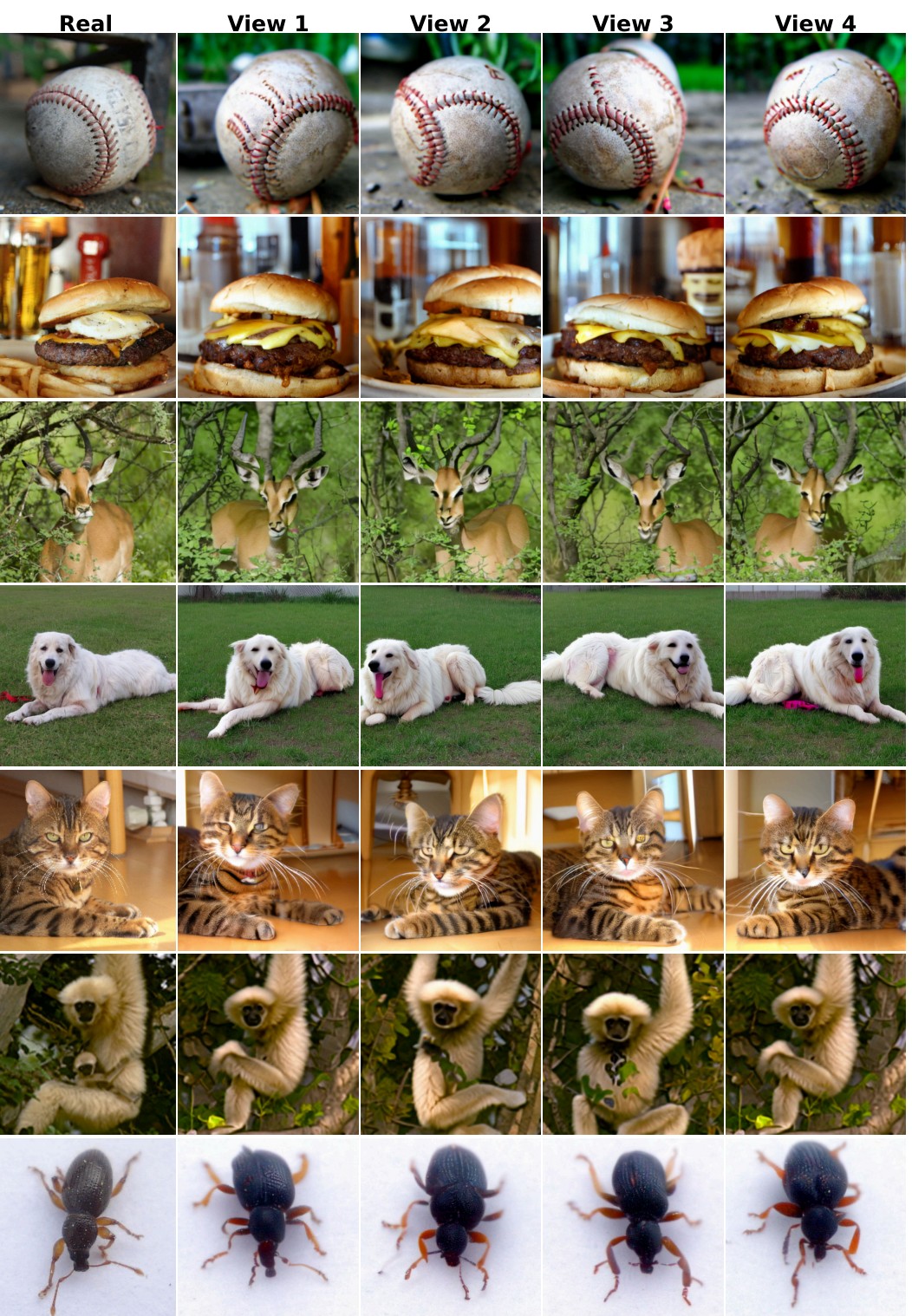

Figure 4: Bag of views visualization. Real denotes an image in ImageNet dataset. The views represent the synthetic views generated with diffusion.

Table 7: Training hyperparameters for CDG-MAE and CropMAE (Eymaël et al., 2024)

| | CDG-MAE (ours) | CropMAE |
|---|---|---|
| Optimizer | AdamW ($\beta_1$=0.9, $\beta_2$=0.95 ) | AdamW ($\beta_1$=0.9, $\beta_2$=0.95 ) |
| Weight decay | 0.05 | 0.05 |
| Base learning rate | $1.5 \times 10^{-4}$ | $1.5 \times 10^{-4}$ |
| Target masking ratio | 90% | 98.5% |
| $lr$ schedule | Cosine Decay | Cosine Decay |
| Epochs | 100 | 100 |
| Batch size | 2048 | 2048 |
| Bag of views size ($M$) | 4 | – |
| Augmentations | Crop [0.5, 1.0] | LocalToGlobal Crop [(0.3, 6.0) (0.1, 1.0)] |
| Aspect ratio | (0.75, 1.33) | (0.75, 1.33) |

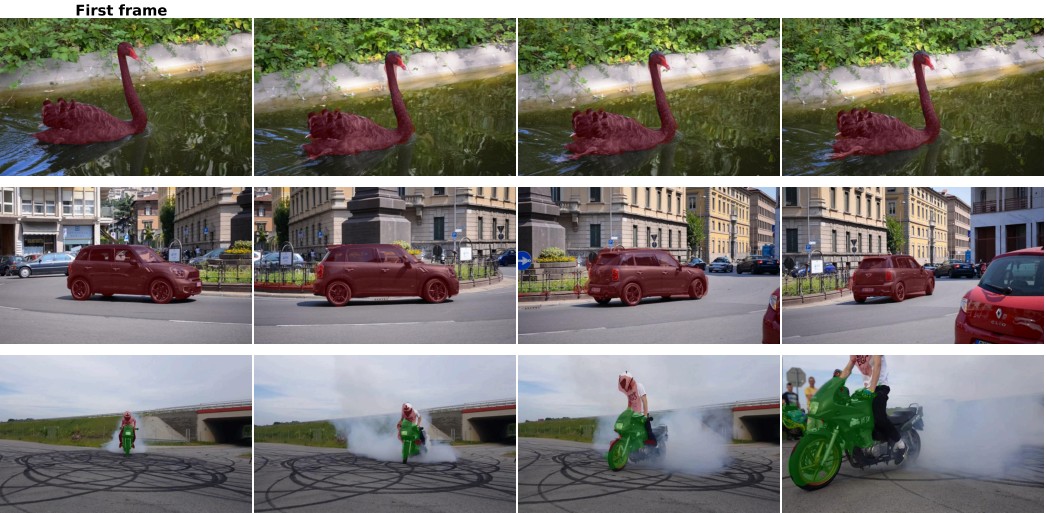

Figure 5: Visualization of label propagation using CDG-MAE ViT-S/16 on DAVIS Pont-Tuset et al. (2017a) dataset. The first frame is annotated with the ground truth object segmentation masks.

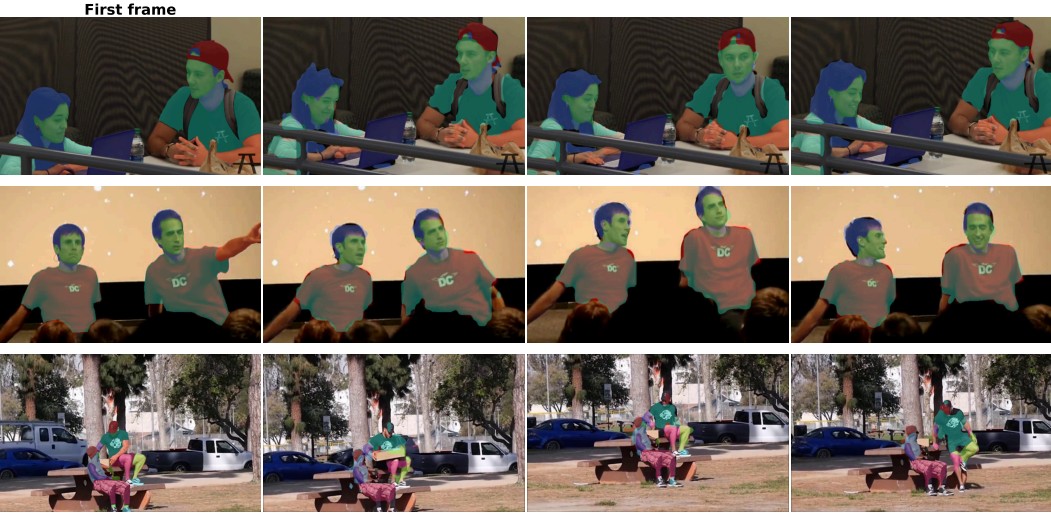

Figure 6: Visualization of label propagation using CDG-MAE ViT-S/16 on VIP Zhou et al. (2018a) dataset. The first frame is annotated with the semantic part segmentation masks.

**First frame**

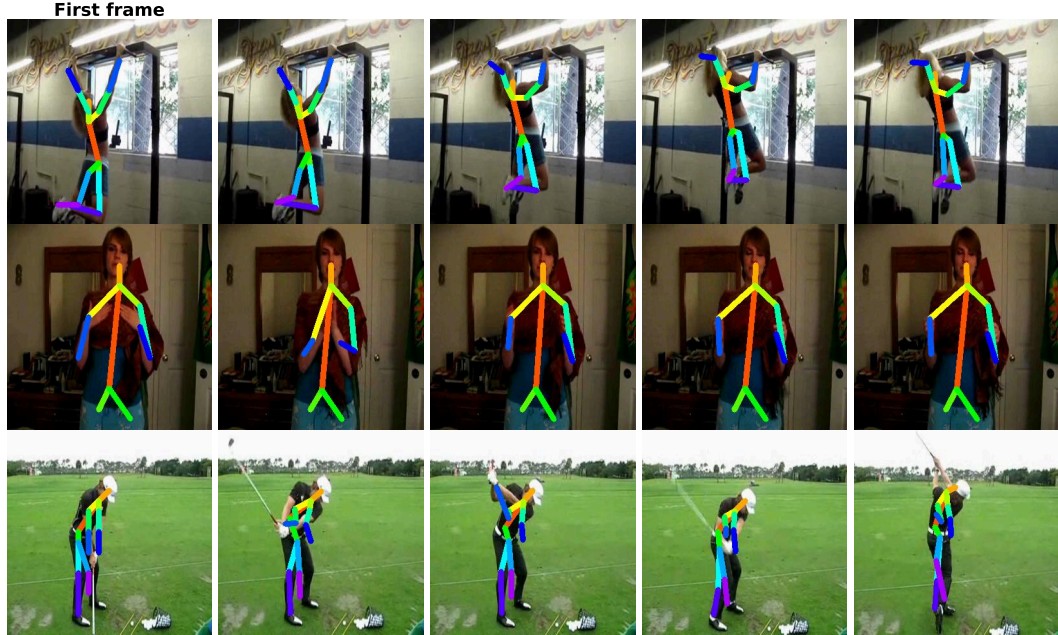

Figure 7: Visualization of label propagation using CDG-MAE ViT-S/16 on JHMDB Jhuang et al. (2013a) dataset. The first frame is annotated with the pose labels.

### A.1.2 DOWNSTREAM EVALUATION

We evaluate our method on standard video label propagation tasks, following the evaluation protocol of previous works (Eymaël et al., 2024; Gupta et al., 2023). We use three datasets: 1) DAVIS (Pont-Tuset et al., 2017a) for video object segmentation, 2) VIP (Zhou et al., 2018a) for semantic part propagation, and 3) JHMDB (Jhuang et al., 2013a) for human pose propagation. In these tasks, annotation is provided for the first frame, and the objective is to propagate ground-truth labels to all subsequent frames of the video.

The evaluation is performed in a training-free manner using $k$-nearest neighbor ($k$-NN) inference. Furthermore, this protocol utilizes a memory queue of the last few frames and restricts source patches to the query's spatial neighborhood. The specific hyperparameter values for this setup are detailed in Table 8.

For DAVIS, we report mean region similarity ($\mathcal{J}_m$), mean contour accuracy ($\mathcal{F}_m$), and their combined average ($\mathcal{J}\&\mathcal{F}_m$). For VIP, we report the mean Intersection over Union (mIoU). For JHMDB, evaluation is based on PCK0.1 and PCK0.2, which represent the percentage of keypoints correctly localized within an error margin of 10% and 20% of the bounding box size, respectively. We use the evaluation codebase released by CropMAE (Eymaël et al., 2024).

Table 8: Hyperparameters for downstream evaluation using $k$-nearest neighbor ($k$-NN) inference.

|                   | DAVIS | VIP | JHMDB |
| ----------------- | ----- | --- | ----- |
| Top-K             | 7     | 10  | 7     |
| Queue Length      | 20    | 20  | 20    |
| Neighborhood Size | 20    | 20  | 20    |

### A.2 VISUALIZATION

Figure 4 shows samples from the bag of views generated using the ImageNet-1K dataset. As seen in the figure, the views exhibit changes in pose, motion, and perspective. Moreover, one can observe that the generated images maintain the main characteristics of the image (objects and background) making them ideal for the training of our CDG-MAE method.

Qualitative results of CDG-MAE on downstream tasks are presented in Figure 5, Figure 6, and Figure 7.

### A.3 COMPUTATION ANALYSIS

#### A.3.1 MULTI-ANCHOR AND ANCHOR MASKING

Table 5 in main paper shows that using multiple anchors with anchor masking improves downstream task performance. In this section and Table 9, we discuss the associated training computational complexity (evaluation-time inference complexity is the same for all models). GLOPs are calculated using a full forward pass up to the loss calculation with a single input training sample. Adding more anchors without anchor masking increases the computational complexity as it results in more tokens being processed by the encoder and decoder. However, introducing anchor masking not only improves performance, but can also help in reducing GLOPs. This is since the masked tokens are dropped at the input and hence do not add complexity, effectively decreasing the number of tokens processed by the encoder for each anchor, and finally the number of tokens used in cross-attention by the decoder. As presented in Table 9, $N = 2$ anchors and $r_a = 50\%$ can maintain almost the same number of FLOPs as the single-anchor setting and outperforms it in downstream tasks. Further, to make the most out of multiple anchors, our best model uses $N = 3$ anchors and $r_a = 25\%$. Finally, for $N = 4$ we noticed no further improvement and even a small decrease in performance. For this reason, we performed only one seed run for $N = 4$.

Table 9: Extension of Table 5 in main paper with GLOPs: Effect of multiple anchors and anchor masking ($r_a$). Multi-anchor training improves performance, and anchor masking offers control over pretext task difficulty, along with reducing training-time computational complexity.

| Num. of Anchors ($N$) | Anchor Masking ratio ($r_a$) | DAVIS $\mathcal{J}\&\mathcal{F}_m$ | VIP mIoU | JHMDB PCK0.1 | GFLOPs |
|---|---|---|---|---|---|
| 1 | 0 | 61.2±0.0 | 37.6 ±0.4 | 46.5 ±0.3 | 6.0 |
| 2 | 0 | 62.0 ±0.1 | 37.6 ±0.1 | 47.1 ±0.2 | 10.4 |
| 2 | 25% | 62.4 ±0.2 | 38.0 ±0.3 | 47.3 ±0.1 | 8.3 |
| 2 | 50% | 62.1 ±0.1 | 38.1 ±0.2 | 47.8 ±0.1 | 6.1 |
| 3 | 25% | 62.6 ±0.1 | 38.1 ±0.1 | 47.8 ±0.2 | 11.6 |
| 3 | 50% | 62.0 ±0.4 | 37.4 ±0.3 | 47.5 ±0.2 | 8.3 |
| 4 | 25% | 62.3 | 37.6 | 47.6 | 14.9 |
| 4 | 50% | 61.7 | 37.4 | 47.6 | 10.6 |

#### A.3.2 BAG OF VIEWS

For each training image in ImageNet-1K (1.28 M images), we generate 4 synthetic views and store them on disk. The total computation cost for the creation of bag of views on a single node of 8 A100 GPUs is approximately 48 hrs and it is performed only once.

#### A.3.3 MACHINE DETAILS AND TOTAL BUDGET

We use a single node with 8 A100 40 GB GPUs for all the experiments. The CDG-MAE ViT-S/16 model takes a maximum of 14 hours for training. A total of 900 node hours were used for this paper, including initial exploration and failed experiments.

### A.4 ADDITIONAL EXPERIMENTS

#### A.4.1 TRAINING CROPMAE FOR MORE EPOCHS

We compare the performance of CDG-MAE trained for 100 epochs with CropMAE trained for a longer schedule (400 epochs). As studied by CropMAE (Eymaël et al., 2024) paper, extended training might lead to saturation in performance. In Figure 8, we observe a similar trend, where the performance of CropMAE starts to decrease on DAVIS and VIP, and saturates on the JHMDB dataset. CDG-MAE trained with 100 epochs outperforms CropMAE even when the latter is trained for longer.

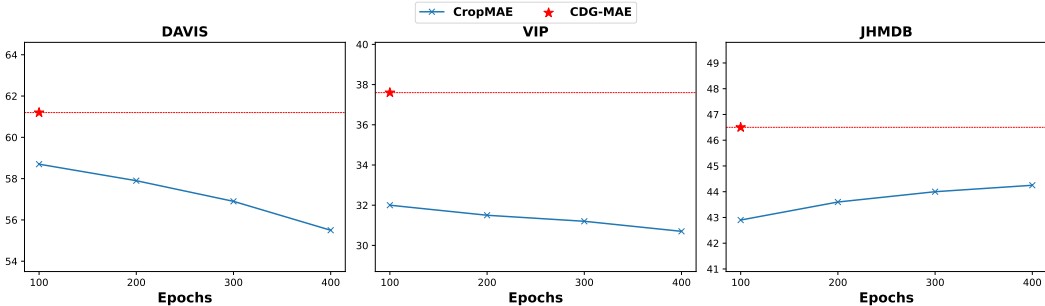

Figure 8: Performance of CropMAE when trained with 400 epochs schedule. We report the evaluation on DAVIS ($\mathcal{J}\&\mathcal{F}_m$), VIP (mIoU), and JHMDB (PCK0.1). We also present the performance of CDG-MAE trained for 100 epochs with single anchor setting.

### A.4.2 TRAINING CROPMAE WITH SYNTHETIC DATA

CropMAE is trained with real images from ImageNet-1K, whereas CDG-MAE is trained with real and diffusion-generated images from ImageNet-1K as pairs of views. In Table 10, we present the performance of CropMAE when trained using real and synthetic images as individual data points (Real + Synthetic). Our proposed CDG-MAE outperforms CropMAE (Real + Synthetic) across all three downstream tasks.

Table 10: Downstream evaluation of CropMAE, CropMAE (Real+Synthetic), and CDG-MAE.

| Method | DAVIS $\mathcal{J}\&\mathcal{F}_m$ | VIP mIoU | JHMDB PCK0.1 |
|---|---|---|---|
| CropMAE | 59.7 | 33.8 | 43.9 |
| CropMAE (Real + Synthetic) | 60.6 | 33.9 | 44.3 |
| CDG-MAE a1 | 61.2 | 37.6 | 46.5 |
| CDG-MAE a3 | 62.6 | 38.1 | 47.8 |

### A.4.3 SCALING

In this section, we compare scaling the number of patches and model parameters in cross-view MAE. In the main paper section 5.5 (also presented in Table 11), we studied scaling the number of patches from 196 to 784 by decreasing the patch size from 16 to 8. For CDG-MAE, scaling from ViT-S/16 to ViT-S/8 led to large improvements across all three tasks (5.8 on DAVIS, 5.2 on VIP, 9.1 on JHMDB). At ViT-S/8, CDG-MAE outperforms image-based CropMAE and closes the gap to the video-based SiamMAE. We believe that for the downstream tasks of video label propagation, scaling the number of patches has more impact than scaling the number of model parameters. Scaling the number of patches helps models learn features for correspondences under fine-grained changes. We observe that CropMAE does not consistently improve when scaling from ViT-S/16 to ViT-S/8. This is likely because cropped views lack sufficient fine-grained variations to benefit training under a smaller patch size. On the other hand, CDG-MAE and SiamMAE improve since they can learn correspondences under fine-grained changes (pose, motion, viewpoint) between diffusion-generated views (in CDG-MAE) and video frames (in SiamMAE).

We conduct an additional experiment by scaling CDG-MAE from ViT-S/16 to ViT-B/16 and observe that it does not lead to consistent improvements across downstream tasks (Table 11). We also report ViT-B/16 performance for SiamMAE and CropMAE. The scaling follows a similar pattern to CDG-MAE on DAVIS and JHMDB. Even for single-image MAE (results of single-image MAE ViT-B/16 and MAE ViT-L/16), performance improves with scale on DAVIS and VIP, but saturates on JHMDB. It is worth noticing that our ViT-S/16 with CDG-MAE performs better than MAE ViT-L/16. Furthermore, previous work addressing cross-view MAE (Gupta et al., 2023; Eymaël et al., 2024), either image- or video-based does not showcase clear and consistent scaling trends in terms of numbers of parameters on video label propagation tasks, which appears to be confirmed in our work.

We believe scaling parameters of cross-view MAE methods remains an open research question. It is still not clear whether it comes from data diversity, downstream tasks or, as we also suspect, from additional required tuning (e.g. hyperparameters) to scale such methods.

Table 11: Scaling number of patches is more effective than scaling number of parameters. † denotes results from our reproduction. ‡ denotes results reported from respective papers.

| Method | Pretraining Data | Arch | DAVIS $\mathcal{J}\&\mathcal{F}_m$ | VIP mIoU | JHMDB PCK0.1 |
|---|---|---|---|---|---|
| CDG-MAE a3 | ImageNet | ViT-S/16 | 62.6 | 38.1 | 47.8 |
| CDG-MAE a3 | ImageNet | ViT-B/16 | 63.3 | 37.6 | 48.0 |
| CDG-MAE a3 | ImageNet | ViT-S/8 | 68.4 | 43.3 | 56.9 |
| CropMAE† | ImageNet | ViT-S/16 | 59.7 | 33.8 | 43.9 |
| CropMAE† | ImageNet | ViT-S/8 | 58.7 | 32.2 | 55.6 |
| CropMAE‡ | ImageNet | ViT-S/16 | 60.4 | 33.3 | 43.6 |
| CropMAE‡ | ImageNet | ViT-B/16 | 60.9 | 32.8 | 44.3 |
| SiamMAE‡ | Kinetics | ViT-S/16 | 62.0 | 37.3 | 47.0 |
| SiamMAE‡ | Kinetics | ViT-B/16 | 62.8 | 38.4 | 47.2 |
| SiamMAE‡ | Kinetics | ViT-S/8 | 71.4 | 45.9 | 61.9 |
| MAE | ImageNet | ViT-B/16 | 53.5 | 28.1 | 44.6 |
| MAE | ImageNet | ViT-L/16 | 56.9 | 29.9 | 44.6 |

### A.4.4 VIDEO DIFFUSION MODELS

We did not experiment with video diffusion models for view generation because they are computationally impractical given our compute budget. SoTA image-to-video models like Wan2.1 (Wan et al., 2025) and HunyuanVideo (Kong et al., 2024) have substantial computational requirements that make them prohibitively expensive for ImageNet-scale data generation (1.3 M images). Wan2.1's 14B model requires 9 minutes to generate a 5 second video on a single A100 GPU. For ImageNet-scale generation, this would require over 1000 days on our 8 * A100 GPU node. Even the more efficient LTX-video (HaCohen et al., 2024) model requires 30 seconds to generate a single video, which would translate to 56 days for ImageNet. However, as video diffusion models become more efficient, we believe they can be easily integrated into our framework. The frames generated by image-to-video diffusion models can be used as a drop-in replacement for the current views generated by image-to-image diffusion models when training CDG-MAE.

### A.5 LIMITATIONS AND FUTURE WORK

Although self-supervised diffusion models can generate diverse variations needed for correspondence learning, we cannot control which specific variations occur between generated views. Future work can study how to better control variations, e.g. pose changes, in the generated views. This direction is challenging and interesting to explore as this should be done in a self-supervised way, i.e. without using pose labels.

