# OpenReview forum: "CDG-MAE: Learning Correspondences from Diffusion Generated Views"
_ICLR.cc/2026/Conference — Submitted to ICLR 2026_

### Official Review · Reviewer_9MqV · 2025-10-30

**Soundness:** 3
**Presentation:** 3
**Contribution:** 2
**Rating:** 4
**Confidence:** 3

**Summary:**

This paper introduces CDG-MAE, a novel self-supervised pre-training method that addresses the data bottleneck in cross-view correspondence learning. The core innovation is leveraging views generated by an image-conditioned diffusion model to synthesize the rich pose and perspective variations typically only found in videos. Furthermore, the paper introduces quantitative consistency metrics (GS, LS, NPS) to guide the selection of the generative model.

**Strengths:**

The central idea of replacing real video frames with synthesized views from a diffusion model is a high-leverage contribution.

The introduction of quantitative Consistency Metrics (GS, LS, NPS) to empirically validate the utility of different diffusion models for correspondence learning is valuable.

The method outperforms its image-based baseline, CropMAE, confirming that diffusion-generated data provides the necessary fine-grained variations that cropping lacks.

**Weaknesses:**

The primary weakness lies in the dependence on a 2D image-to-image diffusion model for view generation. This generative process does not enforce multi-view geometric or topological consistency in 3D space. Specifically, in cases where new objects or details (e.g., the number of branches behind the monkey) should appear or disappear due to viewpoint change, the diffusion model may randomly hallucinate, add, or delete structural elements. This creates training pairs where no true 3D correspondence ground truth exists, forcing the network to fit "generative noise" rather than robust physical correspondence.

In addition, the proposed Consistency Metrics (GS/LS/NPS) are proxy metrics calculated in a feature space. They measure feature similarity but do not guarantee geometric fidelity. The strong correlation observed might merely indicate the generated views share the semantic style of real video frames, not their structural integrity.

**Questions:**

While the features learned by CDG-MAE excel in pixel-level label propagation and show robustness to large temporal shifts, a question remains regarding the method's necessity compared to established geometric and hand-crafted feature methods (e.g., SIFT/RANSAC, dedicated optical flow networks like RAFT, or even SOTA 3D foundation models, e.g. Mat3R, VGGT, etc.）

---

> ### Author Response · Authors · 2025-12-03
>
> We respond to the weaknesses (W) and questions (Q) raised by the reviewer below:
>
> **W1**: The primary weakness lies in the dependence on a 2D image-to-image diffusion model for view generation. This generative process does not enforce multi-view geometric or topological consistency in 3D space. Specifically, in cases where new objects or details (e.g., the number of branches behind the monkey) should appear or disappear due to viewpoint change, the diffusion model may randomly hallucinate, add, or delete structural elements. This creates training pairs where no true 3D correspondence ground truth exists, forcing the network to fit "generative noise" rather than robust physical correspondence.
>
> **Response**: Please refer to the common comment.
>
>
> **W2**: In addition, the proposed Consistency Metrics (GS/LS/NPS) are proxy metrics calculated in a feature space. They measure feature similarity but do not guarantee geometric fidelity. The strong correlation observed might merely indicate the generated views share the semantic style of real video frames, not their structural integrity.
>
> **Response**: We do not claim that the proposed metrics guarantee geometric fidelity; however, we argue that they ensure semantic consistency. As evidenced by CDG-MAE’s superior performance, this consistency is sufficient for video label propagation tasks. We reiterate that our method is designed for video label propagation, consistent with prior works such as CropMAE [1] and SiamMAE [2].
>
>
> **Q1**: While the features learned by CDG-MAE excel in pixel-level label propagation and show robustness to large temporal shifts, a question remains regarding the method's necessity compared to established geometric and hand-crafted feature methods (e.g., SIFT/RANSAC, dedicated optical flow networks like RAFT, or even SOTA 3D foundation models, e.g. Mat3R, VGGT, etc.）
>
> **Response**:  Regarding the models (RAFT, Mat3R, VGGT) suggested by the reviewer, we wish to highlight that they are supervised methods, whereas our approach utilizes self-supervised pretraining. We believe that our model can be effectively integrated as a feature extractor to support the supervised training of these models. For example, WAFT [3] (a recent variant of RAFT-style models) requires features from two frames to train the optical flow head. In this context, we incorporate CGD-MAE and CropMAE (our main baseline) for feature extraction and train WAFT.
> We conducted a limited training of 50K iterations on the FlyingChairs [4] dataset and 50K iterations on the FlyingThings [5] dataset, and then evaluated the models on KITTI [6] following WAFT’s zero-shot setting. The results show that CDG-MAE outperformed CropMAE, demonstrating the effectiveness of pretraining with diffusion views over just image crops. We believe the performance of both models can be significantly improved by conducting WAFT’s full-scale training of 50K iterations on FlyingChairs and 100K iterations on FlyingThings; we will include these full-scale results in the camera-ready version.
>
>
>
> | | Endpoint-error (&#8595;)| F1 (&#8595;)|
> |--------------|------------| ------- |
> | CropMAE + WAFT |  4.65 | 15.36 |
> | CDG-MAE + WAFT  |  4.29 |14.45 |
>
> [1] Eymaël, Alexandre, et al. "Efficient image pre-training with siamese cropped masked autoencoders." ECCV, 2024.
>
> [2] Gupta, Agrim, et al. "Siamese masked autoencoders." NeurIPS, 2023.
>
> [3] Wang, Yihan, and Jia Deng. "WAFT: Warping-Alone Field Transforms for Optical Flow." arXiv preprint arXiv:2506.21526 (2025).
>
> [4] Dosovitskiy, Alexey, et al. "Flownet: Learning optical flow with convolutional networks." ICCV 2015.
>
> [5] Mayer, Nikolaus, et al. "A large dataset to train convolutional networks for disparity, optical flow, and scene flow estimation." CVPR 2016.
>
> [6] Geiger, Andreas, et al. "Vision meets robotics: The kitti dataset." The international journal of robotics research 2013.

---

### Official Review · Reviewer_ECfG · 2025-10-30

**Soundness:** 1
**Presentation:** 2
**Contribution:** 1
**Rating:** 2
**Confidence:** 3

**Summary:**

The paper proposes CDG-MAE, a self-supervised cross-view learning framework that leverages diffusion-generated views. The method uses a self-supervised image-conditioned diffusion model to generate synthetic views with pose and perspective variations from static images, and introduces three quantitative consistency metrics (GS, LS, NPS) to assess the quality of these views. A multi-anchor masking strategy is incorporated into the cross-view MAE framework to enhance correspondence learning. The model is trained from scratch on ImageNet-1K and evaluated on DAVIS, VIP, and JHMDB label propagation benchmarks, showing performance superior to CropMAE and approaching that of video-based SiamMAE.

**Strengths:**

- The paper explores the use of diffusion models to generate cross-view data for self-supervised MAE training, offering an alternative way to learn view-consistent representations from static images.
- The study includes systematic ablations on diffusion model choice, number of anchors, masking ratios, and patch sizes, with consistent results and clear performance trends.
- The method achieves performance close to video-based models when trained only on static images, showing feasibility under constrained conditions.

**Weaknesses:**

- **Outdated motivation:** The central premise, that using video data is costly, is no longer convincing given the availability of large-scale open video datasets and efficient video generation models (e.g., Cosmos, HunyuanVideo, Wan). The motivation therefore is outdated and lacks contemporary relevance.
- The image diffusion-generated views are uncontrolled and may not preserve true viewpoint or structural consistency. As a result, the model primarily learns perceptual similarity rather than genuine dense correspondences.
- **Outdated training setup:** The model is trained from scratch on ImageNet-1K using a ViT-S/16 backbone, without leveraging strong existing off-the-shelf visual models such as DINOv3 or SigLIP2. This training design is misaligned with current practices in vision learning.
- All experiments are restricted to classical label propagation benchmarks (DAVIS, VIP, JHMDB), without evaluation on broader correspondence or geometry-related tasks such as optical flow or depth consistency.
- The claim that diffusion-generated views can replace videos holds only under a weak baseline setting and has not been validated with stronger pretrained encoders, making the conclusion less broadly meaningful.

**Questions:**

Could the authors provide additional results using a recent pretrained backbone (e.g., DINOv3 or other 2025-era vision foundation models) to verify whether the proposed diffusion-generated view training still provides benefits under a stronger initialization?

---

> ### Author Response · Authors · 2025-12-03
>
> We respond to the weaknesses (W) and questions (Q) raised by the reviewer below:
>
> **W1**: Outdated motivation: The central premise, that using video data is costly, is no longer convincing given the availability of large-scale open video datasets and efficient video generation models (e.g., Cosmos, HunyuanVideo, Wan). The motivation therefore is outdated and lacks contemporary relevance.
>
> **Response**:
> Our goal with this method is to provide a way to efficiently train from image-only data without requiring access to video data. It is true that large-scale video datasets are available, but we respectfully believe that this does not mean that collecting video data is not costly. Indeed, for specific applications, such as medical imaging (e.g., endoscopic images [1], or skin cancer images), video data might not be available or costly to collect. We do not claim to replace video data, but instead provide a more efficient way to use image-only datasets, and believe that our method could be applied to video. This will be studied in future work.
>
> Regarding the use of video generation models, we believe that they are still expensive to use nowadays. SoTA image-to-video models like Wan2.1 [2] and HunyuanVideo [3] have substantial computational requirements that make them prohibitively expensive for ImageNet-scale data generation. Wan2.1’s 14B model requires ~9 minutes to generate a 5-second video on an A100 GPU. For ImageNet-scale generation (1.3 M images), this would require over 1000 days on an 8 * A100 GPU node. Even the more efficient LTX-video [4] model requires 30 seconds to generate a video, which would translate to 56 days for ImageNet scale generation. Whereas image2image diffusion model used in our experiment requires 12 hours to generate an ImageNet-scale dataset.
>
>
> **W2**: The image diffusion-generated views are uncontrolled and may not preserve true viewpoint or structural consistency. As a result, the model primarily learns perceptual similarity rather than genuine dense correspondences.
>
> **Response**:
> Please refer to the common comment.
>
> **W3**: Outdated training setup: The model is trained from scratch on ImageNet-1K using a ViT-S/16 backbone, without leveraging strong existing off-the-shelf visual models such as DINOv3 or SigLIP2. This training design is misaligned with current practices in vision learning.
>
> **Response**:
> We understand the reviewer's point. While we trained our model from scratch to align with SSL conventions [5,6,7,8] and ensure fair comparison, we agree that applying CDG-MAE to strong pre-trained models is an interesting direction.
>
> Consequently, we trained an additional layer on top of DINOv3 [9] (ViT-S/16 distilled from the 7B model) using CDG-MAE. We observed that pre-trained DINOv3 outperforms CDG-MAE trained from scratch on two out of three datasets, which is expected given its large-scale training (1.7B images, 7B model distillation). However, when DINOv3 is further optimized with CDG-MAE, performance improves across all datasets. This confirms our method's ability to boost the performance of large-scale pre-trained vision encoders.
>
> Our method is of general interest for the community, not just because it improves on powerful models like DINOv3, but its significance is higher in cases where pretrained DINOv3 is not applicable or effective (out-of-domain - medical imaging), and in-domain pretraining with a limited amount of data available.
>
>
> |  | DAVIS      | VIP        | JHMDB      |
> |--------------|------------|------------|--------|
> | CDG-MAE (scratch)         |  62.6 | 38.1 | 47.8 |
> | DINOv3  |  69.7 | 42.5 | 45.2 |
> | DINOv3 + CDG-MAE  |  70.2 | 43.3 | 46.2 |

---

> ### Author Response · Authors · 2025-12-03
>
> **W4**: All experiments are restricted to classical label propagation benchmarks (DAVIS, VIP, JHMDB), without evaluation on broader correspondence or geometry-related tasks such as optical flow or depth consistency.
>
> **Response**: To ensure a fair comparison with previous work (CropMAE[5] and SiamMAE [6]), we evaluated our method on the same downstream tasks. We developed our method with the scope of applicability to video label propagation tasks. While the application of CDG-MAE for geometry-related tasks is interesting, it is beyond the scope of this work.
>
> Following the reviewer's suggestion, we did a preliminary exploration of the use of features from CDG-MAE and CropMAE (our main baseline) for optical flow by training WAFT [10]. We conducted a limited training of 50K iterations on the FlyingChairs [11] dataset and 50K iterations on the FlyingThings [12] dataset, and then evaluated the models on KITTI [13] following WAFT’s zero-shot setting. The results show that CDG-MAE outperformed CropMAE, demonstrating the effectiveness of pretraining with diffusion views over just image crops. We believe the performance of both models can be significantly improved by conducting WAFT’s full-scale training of 50K iterations on FlyingChairs and 100K iterations on FlyingThings; we will include these full-scale results in the camera-ready version.
>
>
> | | Endpoint-error (&#8595;)| F1 (&#8595;)|
> |--------------|------------| ------- |
> | CropMAE + WAFT |  4.65 | 15.36 |
> | CDG-MAE + WAFT  |  4.29 |14.45 |
>
> We were unable to evaluate SiamMAE, as neither the code nor the checkpoints are publicly available.
>
> **Q1**: The claim that diffusion-generated views can replace videos holds only under a weak baseline setting and has not been validated with stronger pretrained encoders, making the conclusion less broadly meaningful.
>
> **Response**: As previously presented, we studied and showed that diffusion-generated views, along with our training setting are also beneficial to the strong pre-trained DINOv3 model.
>
>
> [1] Bravo, Diego, et al. "Gastrohun an endoscopy dataset of complete systematic screening protocol for the stomach." Scientific Data 12.1 (2025): 102.
>
> [2] Wan, Team, et al. "Wan: Open and advanced large-scale video generative models." arXiv preprint arXiv:2503.20314 (2025).
>
> [3] Kong, Weijie, et al. "Hunyuanvideo: A systematic framework for large video generative models." arXiv preprint arXiv:2412.03603 (2024).
>
> [4] HaCohen, Yoav, et al. "Ltx-video: Realtime video latent diffusion." arXiv preprint arXiv:2501.00103 (2024).
>
> [5] Eymaël, Alexandre, et al. "Efficient image pre-training with siamese cropped masked autoencoders." ECCV, 2024.
>
> [6] Gupta, Agrim, et al. "Siamese masked autoencoders." NeurIPS, 2023.
>
> [7] He, Kaiming, et al. "Masked autoencoders are scalable vision learners." CVPR 2022.
>
> [8] Chen, Ting, et al. "A simple framework for contrastive learning of visual representations." International conference on machine learning. PmLR, 2020.
>
> [9] Siméoni, Oriane, et al. "Dinov3." arXiv preprint arXiv:2508.10104 (2025).
>
> [10] Wang, Yihan, and Jia Deng. "WAFT: Warping-Alone Field Transforms for Optical Flow." arXiv preprint arXiv:2506.21526 (2025).
>
> [11] Dosovitskiy, Alexey, et al. "Flownet: Learning optical flow with convolutional networks." ICCV 2015.
>
> [12] Mayer, Nikolaus, et al. "A large dataset to train convolutional networks for disparity, optical flow, and scene flow estimation." CVPR 2016.
>
> [13] Geiger, Andreas, et al. "Vision meets robotics: The kitti dataset." The international journal of robotics research 2013.

---

### Official Review · Reviewer_b1L9 · 2025-11-01

**Soundness:** 3
**Presentation:** 3
**Contribution:** 3
**Rating:** 6
**Confidence:** 3

**Summary:**

CDG-MAE is a self-supervised learning framework using Masked Autoencoders (MAEs) to learn visual correspondences by leveraging synthetic views produced by image-conditioned diffusion models. The paper addresses the limitations in current self-supervised methods by replacing less-diverse video data with more diverse diffusion-generated views. The paper introduces metrics for evaluating the utility of generated views. The proposed multi-anchor masking strategy enhances MAE training difficulty and effectiveness.

**Strengths:**

- The idea of introducing self-supervision diversity through diffusion-generated images is interesting and addresses well-identified issues of crop and video strategies.
- The proposed multi-anchor and anchor masking techniques are sound and seem to be effective.
- The ablation on the design choices is solid and covers a lot of variables.
- The proposed model achieves the state of the art in most of the metrics, proving the performance claims. The authors show that their approach closes the gap to models trained on video.
- Together with the appendix, the experimental setup is well documented, including all hyperparameters.

**Weaknesses:**

- There is not a lot of discussion on the choice of the diffusion model. The authors have chosen an augmentation model. I wonder if novel view models (e.g. ViewCrafter) were considered. It would be a great comparison, and such an approach could enable control over the camera pose.
- It is not fully clear what the impact of separate components is. You could potentially apply multi-anchor and anchor masking to the CropMAE approach and investigate how that affects the performance.
- I would like to see the results with Lumos, which shows high correlation with video samples in Table 4.
- It is my understanding that the main use of the proposed metrics is assessing the quality/utility of the generated views. This was used as guidance for selecting the diffusion model. Can it be used to select the best examples for the bag of views? Could that improve the performance?

**Questions:**

- What is the expectation of using your approach on the Kinetics dataset? Could you augment the video frames to improve the training?

---

> ### Author Response · Authors · 2025-12-03
>
> We respond to the weaknesses (W) and questions (Q) raised by the reviewer below:
>
> **W1**: There is not a lot of discussion on the choice of the diffusion model. The authors have chosen an augmentation model. I wonder if novel view models (e.g. ViewCrafter) were considered. It would be a great comparison, and such an approach could enable control over the camera pose.
>
> **Response**: The reviewer raises concerns about the limited discussion on the choice of the diffusion model. We want to highlight that we have experimented with 3 image2image diffusion models in our paper. We also provide an analysis of which models are suitable for correspondence learning in Section 5.2 and Table 4.
>
> We thank the reviewer for their suggestions regarding novel view generation models. Below, we provide the reason for not using novel view generation models. While novel view generation models like Viewcrater [1] and Stable Virtual Camera [2] (a more recent model) offer control over camera pose, the generated images lack the necessary dynamics of the objects themselves, leading to static scenes that are not changing (unlike what happens in videos). This dynamic behavior is crucial for learning correspondence in downstream video tasks, such as those used in our experiments (following CropMAE [3] and SiamMAE [4]). Hence, we did not opt to use novel view methods during our submission. During the rebuttal period, we performed preliminary exploration with the best-performing CDG-MAE model tuned on novel views. Using Stable Virtual Camera, we generated 30 novel views for 10,000 randomly sampled ImageNet images (10 per class), a process that took 24 hours for the limited sample set. Unfortunately, we observed a decrease in performance after tuning the model with these generated views, leading us to conclude that the application of novel view models requires further, more in-depth exploration beyond the scope of this paper.
>
> | Model  | DAVIS      | VIP        | JHMDB      |
> |--------------|------------|------------|--------|
> | CDG-MAE          |  62.6 | 38.1 | 47.8 |
> | CDG-MAE (tuned with novel views) |  60.6 | 36.6 | 47.7|
>
>
>
> **W2**: It is not fully clear what the impact of separate components is. You could potentially apply multi-anchor and anchor masking to the CropMAE approach and investigate how that affects the performance.
>
> **Response** The reviewer raises a question on the applicability of our proposed multi-anchor and anchor masking strategy on CropMAE. To address this, we trained CropMAE with the same multi-anchor setting as proposed by our method. We use our 2 best settings (2 anchors, 50% anchor masking, and 3 anchors, 25% anchor masking). As shown in the table below, we observe that using multiple anchors leads to a decrease in performance for CropMAE. This is expected as additional anchors come from the same image, providing redundant information and also making the pretraining task easier, while for our CDG-MAE this strategy is applied in different generated views, highlighting the importance of the integration of generated data.
>
>
> |   | DAVIS      | VIP        | JHMDB      |
> |--------------|------------|------------|--------|
> | CropMAE - 1 anchor          |  59.7 | 33.8 | 43.9 |
> | CropMAE - 2 anchors, 50% anchor masking |  44.8 | 26.3 | 41.8 |
> | CropMAE - 3 anchors, 25% anchor masking  |  36.0 | 21.6 | 40.6 |
>
>
> [1] Yu, Wangbo, et al. "Viewcrafter: Taming video diffusion models for high-fidelity novel view synthesis." T-PAMI 2025.
>
> [2] Zhou, Jensen, et al. "Stable virtual camera: Generative view synthesis with diffusion models." ICCV 2025.
>
> [3] Eymaël, Alexandre, et al. "Efficient image pre-training with siamese cropped masked autoencoders." ECCV, 2024.
>
> [4] Gupta, Agrim, et al. "Siamese masked autoencoders." NeurIPS, 2023.

---

> > ### Author Response · Authors · 2025-12-03
> >
> > **W3**: I would like to see the results with Lumos, which shows high correlation with video samples in Table 4.
> >
> > **Response**: The reviewer is interested in the results of CDG-MAE when trained with views from the LUMOS model. We want to highlight that Table 4 in the main paper already presents the results for CDG-MAE when trained on data generated with LUMOS with a single-anchor setting. The results under the multi-anchor setting are provided below. We observe that multi-anchor and anchor masking boost the performance over a single anchor setting, highlighting the benefits of our proposed multi-anchor strategy across different diffusion models.
> >
> >
> > | Using Lumos  | DAVIS      | VIP        | JHMDB      |
> > |--------------|------------|------------|--------|
> > | CDG-MAE - 1 anchor          |  61.9 | 37.7 | 47.3 |
> > | CDG-MAE - 2 anchors, 50% anchor masking |  62.7 | 38.3 | 47.8 |
> > | CDG-MAE - 3 anchors, 25% anchor masking  |  62.4 | 37.7 | 47.8 |
> >
> > For our main results in the paper, we chose Gen-SIS as our default diffusion model because it was pretrained only on ImageNet. This is unlike Lumos, which was pretrained on a much larger dataset that poses data leakage risks; as we pretrain CDG-MAE only on ImageNet to remain comparable to our main baseline, CropMAE.
> >
> >
> > **W4**:  It is my understanding that the main use of the proposed metrics is assessing the quality/utility of the generated views. This was used as guidance for selecting the diffusion model. Can it be used to select the best examples for the bag of views? Could that improve the performance?
> >
> > **Response**: The reviewer inquired whether our proposed consistency metrics could be used to select the best views from a bag and whether this would improve performance. Indeed, the reviewer is right, and these metrics can be used to select views within the bag; however, we would like to clarify that we did not use consistency metrics to filter views. We observed that the standard deviation of metrics across 4 different views was less than 1%; therefore, filtering or selecting specific views from the bag was not necessary.
> >
> > **Q1**: What is the expectation of using your approach on the Kinetics dataset? Could you augment the video frames to improve the training?
> >
> > **Response**: This is an interesting suggestion that we will consider in future work. Using diffusion models to augment video frames is indeed relevant. We believe the gains from CDG-MAE should transfer from image-only settings to video, as the high-quality dynamics of video frames can be complemented by the diverse views generated by image diffusion models.

---

### Official Review · Reviewer_WBZA · 2025-11-01

**Soundness:** 3
**Presentation:** 3
**Contribution:** 2
**Rating:** 4
**Confidence:** 3

**Summary:**

This paper presents CDG-MAE, a self-supervised framework that combines diffusion-based novel view generation with a multi-anchor masked autoencoder. The method replaces costly video data with diffusion-generated synthetic views that introduce pose and perspective diversity while preserving global consistency. A new multi-anchor masking strategy improves task difficulty and representation robustness. Empirically, CDG-MAE outperforms image-only methods such as CropMAE and narrows the gap to video-based SiamMAE on multiple label-propagation benchmarks.

**Strengths:**

- Creative use of diffusion for correspondence learning, addressing the lack of video data for cross-view pretraining.
- Multi-anchor masking is a well-motivated and effective extension to SiamMAE.
- Comprehensive experiments show consistent gains across three datasets, with strong ablations on masking ratios and diffusion backbones.

**Weaknesses:**

- The technical novelty mainly lies in the proposed consistency metrics (GS–LS–NPS) for selecting diffusion-generated views, but their contribution is not deeply analyzed (e.g., what if LS is omitted, or completely remove this metric or GS alone suffices?).
- Other elements (diffusion-based augmentation, Siamese MAE) are incremental combinations of prior work (Gen-SIS, CropMAE).
- The experimental organization could be improved by presenting the main comparison table earlier.

**Questions:**

- Why can diffusion-generated views, which are often inconsistent, still yield effective correspondences?
- Please provide ablations isolating each consistency metric (GS, LS, NPS).
- Move the main results table earlier for clarity.

---

> ### Author Response · Authors · 2025-12-03
>
> We respond to the weaknesses (W) and questions (Q) raised by the reviewer below:
>
> **W1, Q2** The technical novelty mainly lies in the proposed consistency metrics (GS–LS–NPS) for selecting diffusion-generated views, but their contribution is not deeply analyzed (e.g., what if LS is omitted, or completely removing this metric or GS alone suffices?).
>
>
> **Response**:
> The reviewer raises concerns regarding the limited analysis of consistency metrics. To justify the necessity of the full GS–LS–NPS metric set, we conducted an analysis using static images and included an additional row of computation in our results.  To make the impact of the different metrics clearer, we study a simple edge-case setting where two views are the exact same image (referred to as “Static images” in the Table below). This is, of course, not a realistic scenario, but it serves as a critical diagnostic. Static images are unsuitable for correspondence learning because they lack change between views. Relying only on Global Similarity (GS) or Nearest Patch Similarity (NPS) would incorrectly identify static images as ideal views due to the high values of GS and NPS. The Local Similarity (LS) metric is thus essential because it imposes a high penalty on static images. A high LS score indicates a lack of sufficient structural difference between the views, proving LS is necessary to filter out these uninformative views.
> Furthermore, while GS and NPS can be correlated in terms of measurements, we believe NPS is still an important metric because it computes similarity at the patch level, aligning more directly with the requirements of correspondence learning. In contrast, GS uses a globally pooled representation, making it less sensitive to the crucial local changes that define useful view pairs. Therefore, the complete GS–LS–NPS set is ideal: GS and NPS ensure semantic consistency, while LS ensures the presence of non-trivial, meaningful variation.
>
> Extended version of Table 4 (main paper)
> | Diffusion model | DAVIS | VIP  | JHMDB | Global Sim. - GS (&#8593;) | Local Sim. (LS) (&#8595;) | Nearest Patch Sim. - NPS (&#8593;)  |
> |-----------------|-------|------|-------|------------|-----------|-----------|
> | Gen-SIS         | 61.2  | 37.6 | 46.5  | 0.992      | 0.377     | 0.795     |
> | RCG             | 57.4  | 34.8 | 43.7  | 0.955      | 0.308     | 0.738     |
> | LUMOS           | 61.9  | 37.7 | 47.3  | 0.995      | 0.376     | 0.812     |
> |                 |       |      |       |            |           |           |
> | Video frames |    N/A   |   N/A   |   N/A    | 0.992      | 0.389     | 0.884     |
> | Static images |    N/A   |   N/A   |   N/A    | 1      | 1     | 1     |
>
> **W2**: Other elements (diffusion-based augmentation, Siamese MAE) are incremental combinations of prior work (Gen-SIS, CropMAE).
>
> **Response**: The reviewer raises the concern about the novelty of our work. We respectfully disagree with the reviewer and highlight that our contributions are two-fold: 1) Pioneering the usage of diffusion-generated views for correspondence learning, and 2) Architectural innovation in incorporating multiple anchors and anchor masking into standard MAE training.
>
> We investigate for the first time the use of diffusion-generated views for MAE training, and also propose metrics to evaluate the relevance of the generated images for correspondence learning. With the rising performance of diffusion models and their impressive quality on image generation, we believe that such data could play an important role in pretraining and, in particular, cross-view MAE pretraining. To the best of our knowledge, no prior work has explored this direction. We also show that the integration of diffusion for cross-view into MAE requires careful considerations. Indeed, the underlying characteristics of the diffusion model play an important role. For this reason, we also introduce quantitative metrics (Global, Local, Nearest Patch Similarity) in section 3.2 to evaluate the relevance of diffusion-generated views for correspondence learning. We use these metrics to select the diffusion model in Table 4 (main paper).
>
> Our primary innovation to utilize diffusion-generated views for MAE motivates our main architectural contribution to standard MAE (multi-anchor and anchor masking). Even with a single anchor (CDG-MAE-a1), our method significantly outperforms CropMAE across all tasks (1.5% on DAVIS, 3.8% on VIP, 2.6% on JHMDB - main paper Table 6), demonstrating the independent effectiveness of diffusion-generated views. When using multiple anchors, the performance further improves by 1.4% on DAVIS, 0.5% on VIP, and 1.3% on JHMDB (main paper Table 6).
>
>
> **Q1**: Why can diffusion-generated views, which are often inconsistent, still yield effective correspondences?
>
> **Response**: Please refer to the common comment.

---

### Author Response · Authors · 2025-12-03
**[Common Comment] Diffusion-generated views can yield effective correspondences even though generated views do not guarantee geometric consistency.**

We provide a common response to the following concerns

**Reviewer WBZA** (Q1) asks a question about why diffusion-generated views, which are often inconsistent, still yield effective correspondences?

**Reviewer ECfG** (W2) raises the concern that the image diffusion-generated views are uncontrolled and may not preserve true viewpoint or structural consistency.

**Reviewer 9MqV** (W1) states the primary weakness lies in the dependence on a 2D image-to-image diffusion model for view generation. This generative process does not enforce multi-view geometric or topological consistency in 3D space.

**Response**: We agree with the reviewers that 3D consistency in generated views can be beneficial; however, we argue that 3D consistency is not necessary within the scope of our work. The diffusion-generated views indeed exhibit local geometric inconsistencies, but still yield effective correspondences due to their high semantic consistency and their ability to closely mimic the variations found between natural video frames (as analyzed in Table 1 of our main paper). Semantic consistency means the views preserve the identity and relationships of key objects, which is the essential information required for establishing good correspondence. Since actual video frames are considered the ideal source for learning correspondences (as they naturally capture subtle motion and pose changes), views generated by the diffusion model that replicate these specific variations are also well-suited for the task. We show in Table 1 that diffusion-generated views closely mimic video frames. Moreover, the label-propagation evaluation tasks we consider require learning to find correspondences between frames of a video. We follow the exact downstream task collection of previous works, namely CropMAE [1] and SiamMAE [2], for fair comparison. The boost coming from our method on such tasks thus indicates that correspondences are present in the generated training images.

[1] Eymaël, Alexandre, et al. "Efficient image pre-training with siamese cropped masked autoencoders." ECCV, 2024.

[2] Gupta, Agrim, et al. "Siamese masked autoencoders." NeurIPS, 2023.

---

### Author Response · Authors · 2025-12-03
**Summary of our responses to key concerns**

Below we provide a summary of the key concerns raised by the reviewers and our response.

**DINOv3**: Reviewer ECfG raises a concern that CDG-MAE is trained from scratch and does not utilize existing strong pre-trained models, such as DINOv3. We want to highlight that we trained our model from scratch to align with SSL conventions and ensure a fair comparison with previous works (SiamMAE and CropMAE). However, we agree that applying CDG-MAE to strong pre-trained models is an interesting direction, and we have shown in this rebuttal that CDG-MAE can further boost the performance of the already robust DINOv3 model.

**Geometric inconsistency in diffusion views**: Reviewers WBZA, ECfG, and 9MqV raised the concern that the diffusion-generated views in CDG-MAE lack geometric consistency. While we agree that these views do not guarantee 3D consistency, we argue that this property is not required for our tasks. We demonstrate that diffusion-generated views provide semantic consistency comparable to video frames, which is sufficient for label propagation. This is confirmed by our consistency metrics and CDG-MAE's superior performance over crop-based methods (CropMAE), closing the gap with video-based methods (SiamMAE).

**Optical Flow evaluation**: Reviewers ECfG and 9MqV raised concerns that the current evaluation is restricted to video label propagation tasks and does not include tasks such as optical flow. We would like to emphasize that we followed the same set of downstream tasks as previous works (CropMAE and SiamMAE) for a fair comparison. However, during the rebuttal, we evaluated CropMAE and CDG-MAE for optical flow and demonstrated that CDG-MAE substantially outperforms CropMAE. This indicates the effectiveness of diffusion-generated views in learning better correspondences than the image-crop-only method (CropMAE).

---

### Meta-Review · Area_Chair_s3QD · 2025-12-24

**Summary:**

The paper introduces CDG-MAE, a self-supervised framework that leverages diffusion-generated views to train a Siamese Masked Autoencoder for visual correspondence learning. The core motivation is to bypass the need for costly video data by generating synthetic views with pose and perspective variations from static images. The method introduces specific consistency metrics (GS, LS, NPS) to select appropriate diffusion models and employs a multi-anchor masking strategy.

While the proposed multi-anchor strategy and the exploration of diffusion for SSL are interesting, the consensus among reviewers is that the paper falls below the acceptance threshold (Scores: 2, 4, 4, 6). The primary concerns revolve around the validity of the motivation (given the availability of video data/models), the geometric consistency of 2D diffusion generations, and the incremental nature of the contribution.

**Reviewer Concerns:**

**Addressed Concerns:**

1. Modern Backbones:Reviewer ECfG questioned the use of ViT-S/16 from scratch. In the rebuttal, the authors successfully demonstrated that CDG-MAE can improve the performance of a strong DINOv3 baseline, which strengthens the method's applicability.

2. Consistency Metrics: Reviewer WBZA questioned the necessity of the proposed metrics (GS/LS/NPS). The authors provided clarifications and analysis on static images to justify the inclusion of Local Similarity (LS).

3. Geometric Evaluation: In response to Reviewer ECfG and Reviewer 9MqV asking for geometry-related tasks, the authors provided preliminary results on Optical Flow (WAFT), showing CDG-MAE outperforms CropMAE.

**Outstanding Concerns (Key Reasons for Rejection):**

1. Geometric Consistency & Hallucination (Critical): Reviewer 9MqV and Reviewer ECfG raised a fundamental issue: 2D image-to-image diffusion models do not enforce 3D geometric consistency. They can "hallucinate" or alter topological details (e.g., branches appearing/disappearing) during view generation. While the authors argue that "semantic consistency" is sufficient for label propagation, the reviewers remain unconvinced that learning from non-physical "generative noise" is a robust path for correspondence learning compared to using actual video or 3D-consistent generative models.

2. Motivation: Reviewer ECfG argued that the premise—"video data is costly"—is becoming outdated given large-scale open video datasets. While the authors defended this by citing medical domains, the paper's experiments focus on general vision benchmarks where video is abundant, creating a mismatch between motivation and evaluation.

3. Incremental Novelty: Reviewer WBZA noted that the method largely combines existing components (Gen-SIS/Lumos + CropMAE) with a new masking strategy. The technical novelty of the consistency metrics was viewed as limited.

**Reviewer Scores:**

- Reviewer WBZA (Score: 4): Maintained. Found the technical novelty of the metrics limited and the approach incremental.

- Reviewer b1L9 (Score: 6): Supportive. Appreciated the ablation studies and the rebuttal experiments (DINOv3), but stands alone as the only positive score.

- Reviewer ECfG (Score: 2): Maintained. Strongly questions the motivation (video availability) and the training setup compared to modern standards.

- Reviewer 9MqV (Score: 4): Maintained. Concerns about the lack of geometric fidelity in diffusion-generated views remain the primary bottleneck.

---

### Decision · Program_Chairs · 2026-01-26

Reject